# Antiplatelet Versus Anticoagulation for Asymptomatic Patients with Vertebral Artery Injury During Anterior Cervical Surgery—Two Case Reports and Review of Literature

**DOI:** 10.3390/brainsci9120345

**Published:** 2019-11-28

**Authors:** Michael Hall, David Cheng, Wayne Cheng, Olumide Danisa

**Affiliations:** 1UCR School of Medicine, Riverside, CA 92507, USA; 2Loma Linda Medical Center, Loma Linda, CA 92354, USA; Davidwaynecheng@gmail.com (D.C.); Spinesurgeon1995@gmail.com (W.C.)

**Keywords:** antiplatelets, anticoagulation, vertebral artery injury, cervical surgery

## Abstract

Vertebral Artery Injury (VAI) while performing cervical spinal reconstruction surgery is rare, but it can lead to catastrophic events. Treatment for this injury with regard to antiplatelet versus anticoagulation therapy is controversial. The purpose of this report is to discuss two cases of VAI that occurred during the performance of cervical reconstruction surgery and provide a guideline based on a literature review about whether to use anticoagulant or antiplatelet therapy for treatment of asymptomatic VAI. In case 1, iatrogenic injury occurred to the left C5 vertebral artery (VA) during high speed burr removal of an osteophyte on the left C5/6 uncovertebral joint, resulting in VAI. This patient was treated with Heparin resulting in respiratory complication. Case 2 encountered VAI while using the kerrison rongeur to perform a right sided C5/6 foraminotomy. Antiplatelet therapy was administered. Fourteen publications relevant to Antiplatelet versus Anticoagulation treatment were reviewed. Anticoagulation has similar results to antiplatelet therapy. Studies are limited; there were no common guidelines or parameters concerning the utilization of Antiplatelets versus Anticoagulants. Anticoagulation achieved similar results as Antiplatelet therapy; based on the limited relevant data, the superiority of one treatment over the other cannot be concluded in VAI after cervical spinal reconstruction surgery.

## 1. Introduction

Vertebral Artery Injury (VAI) can be a devastating event due to causing the following—arterial dissection, hematoma formation, aneurysm or vascular occlusion. This in turn can result in fatal ischemic injury or potentially permanent neurological impairment. Unilateral damage to the vertebral artery (VA), fortunately, is rarely deadly due to the contribution of collateral circulation (contralateral VA and the Circle of Willis) [1]. Injury to this artery is common in cervical blunt trauma [2]. During surgical procedures, such as an anterior corpectomy or anterior cervical fusion, the VA remains vulnerable due to its unique anatomical location within the C2–C6 transverse foramina. Various clinical syndromes can manifest from this injury, such as lateral medullary syndrome (Wallenberg Syndrome) [3].

Treatment options for VAI are—observation, anticoagulation or antiplatelet therapy. No preferred method has been proven to be superior [4]. The controversy of anticoagulation versus antiplatelet therapy is a topic of interest in the field of spine surgery. Literature concerning this controversy is limited. In our case presentations, we decided to treat with anticoagulation therapy in one patient, and with antiplatelet therapy in the other.

## 2. Materials and Methods

We report two cases of VAI after anterior cervical surgery at our institution from 2015 to 2017. The radiographic features, presenting symptoms, clinical characteristics, type of management and outcomes of the patients were all studied. A literature review was also conducted to evaluate the current management and treatment of asymptomatic VAI—observation, anticoagulation or antiplatelet therapy.

## 3. Results

### 3.1. Case Reports

#### 3.1.1. Case Illustration 1

Patient is a 78 year-old male presenting with history of cervical myelopathy—frequent falls over the past 12 months, difficulty with bowel and bladder control and complaints of diminished dexterity of the hands.

#### 3.1.2. Imaging

Plain X-rays revealed large anterior osteophytes most prominent from C4–C7 with severe spondylosis. MRI images show spinal cord compression from C4–C7 and myelomalacia at C5–C6, retrolisthesis at C4–C5, and severe multilevel disk degeneration (Figure 1).

#### 3.1.3. Surgical Plan

Stage 1—anterior corpectomy at C5, fibula structural allograft, anterior discectomy at C6–C7 and a C4–C7 anterior fusion instrumentation. Stage 2—posterior C4 to C7 posterior decompressive laminectomies, fusion, instrumentation and bone graft.

#### 3.1.4. Surgical Complication

During Stage I surgery, pulsatile bleeding was encountered when using a high speed burr to remove an osteophyte on the left of C5/6 uncovertebral joint vertebral body. Bone wax was applied to regain hemodynamic control. Four hundred ml of blood was lost but the remainder of the surgery was uneventful.

The patient was asymptomatic immediately post-op. Computed tomography (CT) angiography showed anterolateral bony defect at C5 (Figure 2). The study also revealed 50% focal narrowing of the left VA at C5; consistent with focal thrombus versus external compression possibly due to bone wax (Figure 3). The study confirmed acceptable cervical alignment, operative decompression, and hardware placement.

#### 3.1.5. Case Illustration 1

The patient was admitted to the intensive care unit and Mean Arterial Pressures (MAPs) were maintained at 90 mmHg, and he was fully anticoagulated with a heparin drip. He, however, developed significant airway edema and respiratory distress within 24 h after anticoagulation and hypertensive therapy. The patient thus required emergent intubation and subsequent operative hematoma evacuation. Lateral radiograph 1-year post operation shows C4–C7 fusion hardware and resolution of hemodynamic instability (Figure 4).

#### 3.1.6. Case Illustration 2

The patient is a 57 year-old female with a past medical history (PMH) positive for a pacemaker and an embolic cerebral vascular event (CVA) at age 50. She presented with a 9-month history of cervical radiculopathy manifested by severe neck pain, right shoulder and arm pain, along with mild weakness in the right biceps. Sensory deficits noted in the C5 and C6 dermatomes. She had failed conservative management (physical therapy, chiropractic manipulation, acupuncture) and cervical epidural.

#### 3.1.7. Imaging

Plain radiograph showed multi-level cervical spondylosis. CT myelogram of the cervical spine from an outside institution demonstrated cervical stenosis, worst at the C4/5 and C5/6 levels (Figure 5). Magnetic Resonance Imaging (MRI) was unable to be performed due to lack of insurance authorization.

#### 3.1.8. Surgical Plan

C4/5 and C5/6 anterior cervical discectomy and fusion (ACDF) with central and foraminal decompressions at corresponding levels.

#### 3.1.9. Surgical Complication

During the right sided foraminotomy at C5/6, brisk bleeding was noted while using a kerrison rongeur to remove the medial uncovertebral joint. Hemostatic agents and cottonoids were used in unison to control the bleeding. A total of 500 mL of blood was lost. Despite a brief decrease in blood pressure, hemodynamics was stabilized with rapid crystalloid and albumin infusion. Neuromonitoring signals (transcranial MEPs and SSEPs) were stable throughout. The surgery was completed—interbody allograft cages placed, and 2-level plate implanted from C4–C6.

### 3.2. Case Illustration 2

The patient was extubated and taken to the recovery room. Her blood pressure and pulse rate were within normal range. Her neurologic exam was intact with regard to mentation, coordination and motor function. Sensory deficits were unchanged from pre-operative exam. She was taken to the angiography suite and this revealed occlusion of the right VA (Figure 6). She had a contrast brain CT 48 h later, and no evidence of ischemia or infarct was noted. She subsequently was started on Aspirin (ASA) therapy on post-op day # 3. Serial exams throughout her 5-day hospitalization remained unchanged except for improvement in pre-operative sensory deficit. Follow up angiogram 6 months later showed recanalization of right VA (Figure 7).

## 4. Discussion

Case 1 details a case of respiratory distress and airway complications due to postoperative anticoagulant therapy which necessitated emergent intubation and hematoma evacuation. In this case, we decided to treat with anticoagulant (Heparin) therapy after asymptomatic operative VAI. According to the literature, anticoagulant management post VAI has traditionally led to hemodynamic stabilization [5,6,7].

Case 2 also encountered VAI, this time while using the kerrison rongeur to perform a right sided C5/6 foraminotomy. We decided to treat this patient with delayed antiplatelet therapy due to a lack of postoperative neurologic deficits.

Lunardini et al. [8] published a cervical spine research society (CSRS) cross-sectional study where members were surveyed and the incidence of VAI was reported as 0.07% (111/163,324) of cervical surgeries. The specific surgical procedures were extrapolated—posterior instrumentation of the cervical spine (32.4%), anterior cervical corpectomy (23.4%), and posterior cervical exposure (11.7%), anterior discectomy (9%). Outcomes of the VAI showed no permanent neurologic sequelae in 90% of patients, permanent neurologic sequelae in 5.5%, and death in 4%. It was also concluded that less experienced surgeons (performed fewer than 300 cervical lifetime cases) had a fivefold increased risk of causing an iatrogenic VAI. Guan et al. [9] performed a systematic review of 25 papers reporting VAI injury in 54 patients after anterior cervical surgery for the following conditions: degenerative disease (64%), tumor (14%), and trauma (9%). They concluded that preoperative evaluation (including angiography) and real time radiographic guidance reduced VAI risk. They also cautioned against the use of tamponade as definitive treatment since there is a 48% pseudoaneurysm risk.

In our case series, neither patient was asymptomatic after sustaining iatrogenic VAI. In both cases, we did not feel the need for endovascular repair. The issue, then is how to manage the complications. The use of antiplatelet vs anticoagulation therapy for management of VAI remains a controversial topic, there is no substantive literature displaying superiority of a particular pharmacologic modality.

## 5. Literature Review

### 5.1. Anticoagulation Therapy

General guidelines of asymptomatic VAI management involve observation, anticoagulation or antiplatelet modalities [10]. A case series by Schellinger et al. [11] evaluated 4 patients with VAI. Two (50%) patients suffered ischemic mortality from observation without therapy while 1 (25%) symptomatic patient received Heparin leading to resolution of symptoms and 1 (25%) symptomatic patient received no therapy resulting in symptom resolution [11].

In another case series, Willis et al. [12] evaluated 12 asymptomatic patients with VAI. Nine of 12 (75%) patients suffered vertebral artery occlusion (VAO) only to be treated with observation, none suffered ischemic complications. Three of the 12 (25%) patients had evidence of pseudoaneurysm or dissection successfully treated with systemic Heparin [12].

Biffl et al. [5] found that VAI patients treated with systemic Heparin were associated with improved neurological outcomes and decreased incidence of neurological deterioration from onset injury to discharge. Of the 16 VAI patients treated with systemic heparin within the study, 2 (13%) improved, only 4 (25%) increased to higher injury grade and 10 (63%) remained unchanged [5].

### 5.2. Antiplatelet Therapy

Conversely, in another study by Colella et al. [13], 9 patients with mural carotid injury were treated with antiplatelet therapy resulting in 8 (88%) patients recovering neurologically without deficits. Jang et al. [7] conducted a study to evaluate the management of asymptomatic VAO in 8 patients needing cervical fracture surgery. Five of 8 (62%) patients were treated by observation alone while 2 of 8 (25%) patients were treated with aspirin post-operatively, all making a complete recovery without heparin use [7].

### 5.3. Anticoagulation vs. Antiplatelet Therapy

A randomized control trial by the Cervical Artery in Dissection Stroke Study(CADISS) investigators [14] assessed the efficacy of antiplatelets versus anticoagulants in the treatment of extracranial VA dissection. No statistical difference between the two treatments was appreciated (*p* = 0.2). Of the 126 patients treated with antiplatelet modalities, 3 (2%) experienced death or stroke while 1 patient experienced stroke of the 124 treated with anticoagulant modalities (odds ratio [OR] 0.335, 95% CI 0.006–4.233; *p* = 0.63) [14].

## 6. Conclusions

The overall risk of VAI in cervical surgery is low. The majority of VAI have no permanent neurologic impairment, yet this injury serves as a cautionary tale. At our institution, we are currently considering routine preoperative angiography for high risk cases such as anterior cervical corpectomy. We are also starting to implement intraoperative navigation for such cases. First line medical management for asymptomatic VAI includes—observation, anticoagulation and antiplatelet therapy. The therapeutic management of VAI regarding anticoagulation versus antiplatelet modalities are controversial. Conflicting evidence based on the literature makes establishing a preferred method in the management of asymptomatic VAI difficult. The definitive establishment of anticoagulation over antiplatelet therapy or vice versa as the superior method in asymptomatic VAI treatment, is inconclusive. Due to the increased variability in outcomes of asymptomatic VAI treatment; the controversy between antiplatelet and anticoagulation therapy remains.

## Figures and Tables

**Figure 1 brainsci-09-00345-f001:**
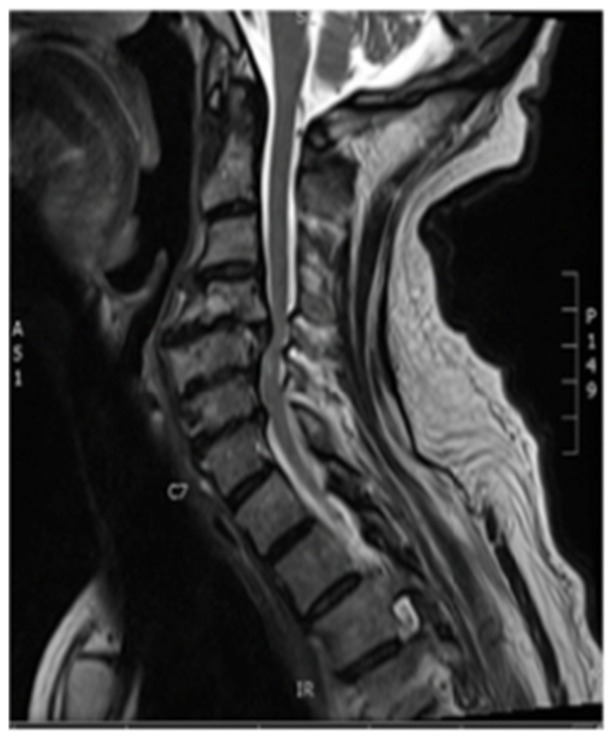
Pre-op Sagittal Magnetic Resonance Image (MRI) (Cervical Spine).

**Figure 2 brainsci-09-00345-f002:**
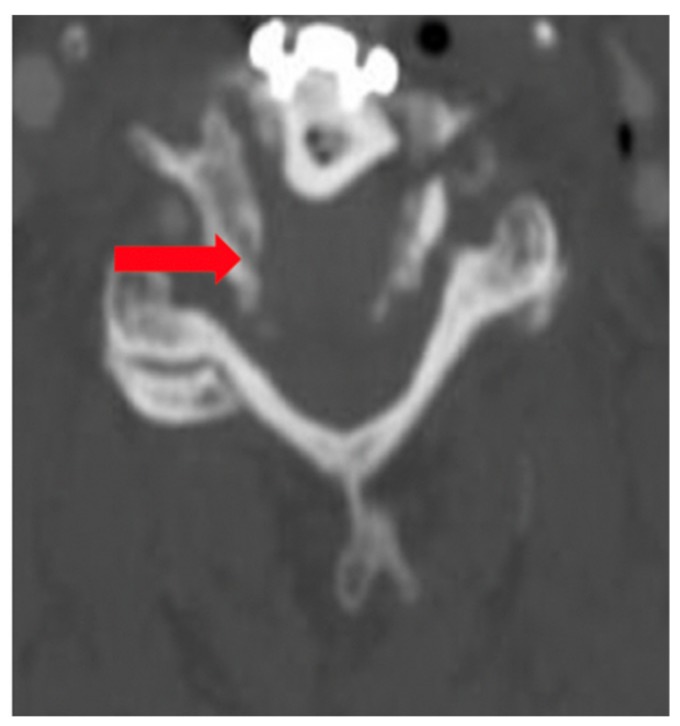
Computed tomography (CT) Scan (arrow denoting bony defect C5).

**Figure 3 brainsci-09-00345-f003:**
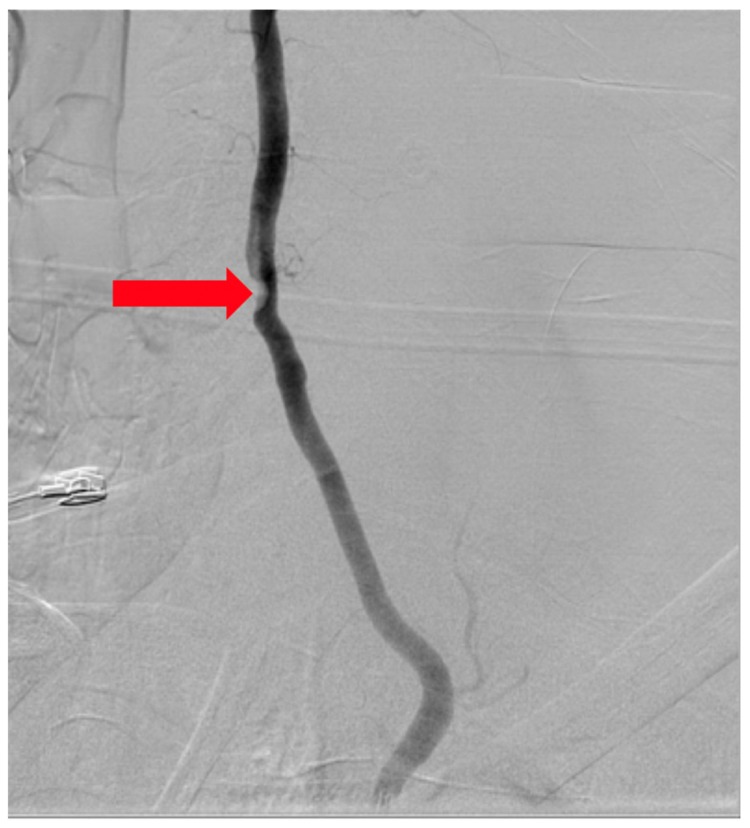
Angiogram (arrow denoting filling defect).

**Figure 4 brainsci-09-00345-f004:**
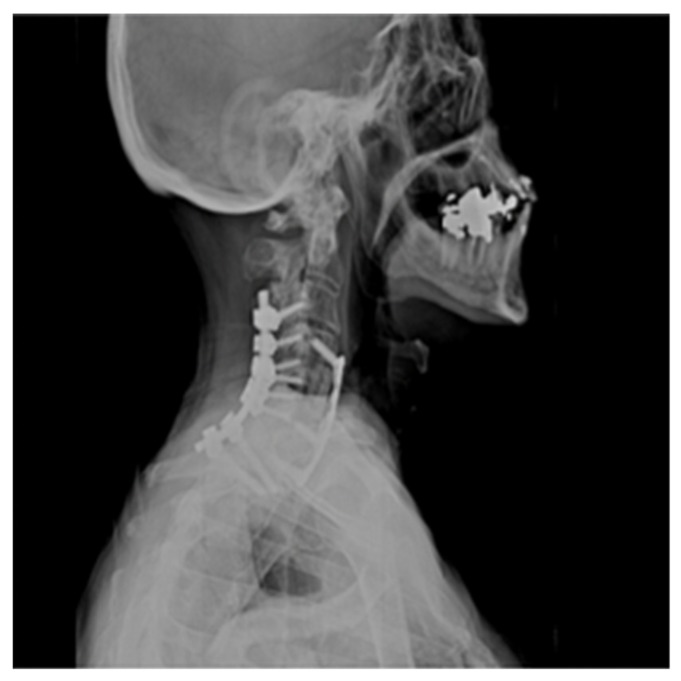
Lateral Radiograph Post Op 1 Year.

**Figure 5 brainsci-09-00345-f005:**
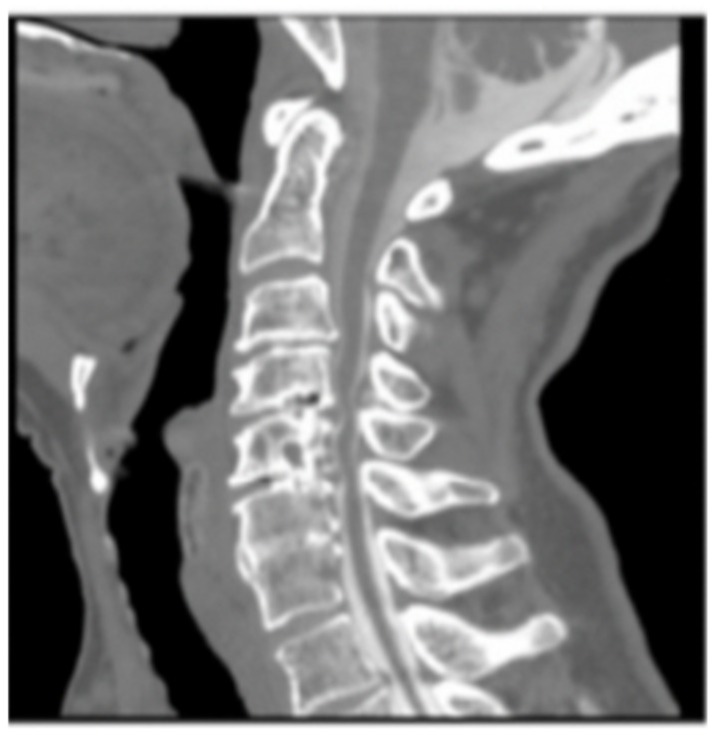
CT myelogram.

**Figure 6 brainsci-09-00345-f006:**
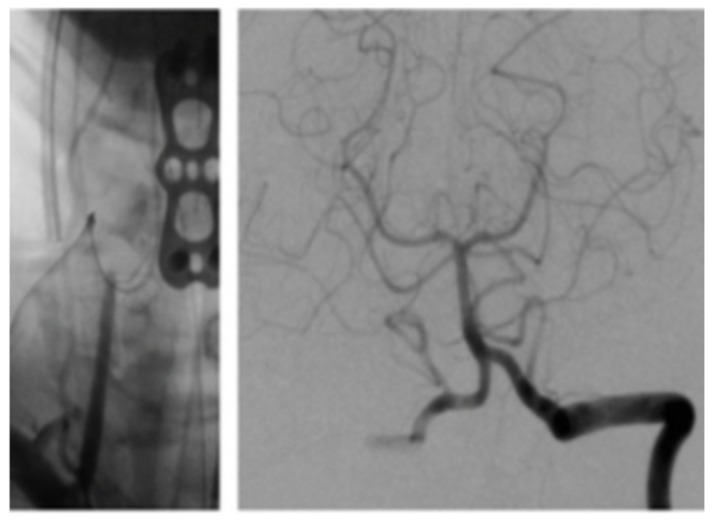
Angiogram post-op.

**Figure 7 brainsci-09-00345-f007:**
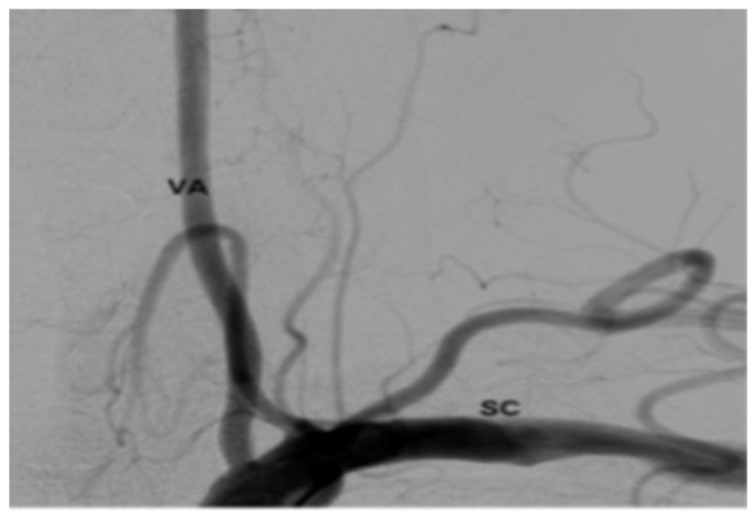
Angiogram 6 months post-op.

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
