# Peer review of "Antiplatelet Versus Anticoagulation for Asymptomatic Patients with Vertebral Artery Injury During Anterior Cervical Surgery—Two Case Reports and Review of Literature"

_brainsci, 2019, doi:10.3390/brainsci9120345_

Round 1

Reviewer 1 Report

The authors present two cases of iatrogenic vertebral artery injury.  Treatment with unfractionated heparin in one case led to a hematoma, which required evacuation.  Treatment with aspirin, in the other case, was associated with a good outcome.  This reviewer has used aspirin, 81 mg daily, beginning on the day of the injury, to manage similar situations, with good results.

The point of the case report is that full anticoagulation with unfractionated heparin can cause a hematoma. This manuscript may be useful as a brief cautionary tale, but it should be reduced substantially in length.   Details about the patient’s original presenting symptoms of spondylosis, for example, are superfluous. The Discussion is an overly long, confused mishmash of references to studies of blunt traumatic cerebrovascular injury and spontaneous cervical arterial dissections, none of which are pertinent to these cases.

Author Response

Response to Reviewer 1 Comments:

Point 1: "The point of the case report is that full anticoagulation with unfractionated heparin can cause a hematoma. This manuscript may be useful as a brief cautionary tale, but it should be reduced substantially in length. Details about the patient’s original presenting symptoms of spondylosis, for example, are superfluous.

Our Response to 1: Thank you for your feedback. We changed the manuscript. We reduced the original presenting symptoms of spondylosis in case illustration 1 in length within the manuscript. (lines 67-70) 

Point 2: "The Discussion is an overly long, confused mishmash of references to studies of blunt traumatic cerebrovascular injury and spontaneous cervical arterial dissections, none of which are pertinent to these cases."

Our response to 2: Thank you for your feedback. We included pertinent and relevant studies within the discussion section ( Lunardini et al, Guan et al, etc) that relate to the 2 cases included within our case studies. ( lines 187-198) 

Reviewer 2 Report

Introduction:

Please start with a more general approach, i.e. on cervical surgery. Afterwards, the mentioned spectrum of complications can be elucidated.

Can the risk of injury of vertebral artery be assessed before the operation (s. Kong et al. 2019: PMID: 31428861; DOI: 10.1007/s00586-019-06111-0).

Methods:

Please state votum of local ethics committee.
"Motor strength was 4 out of 5": Please report the muscle, i.e. M. biceps or triceps brachii, or both.

Results:

Case 1:
Surgical plan: 
Did you use intraoperative navigation systems? If so, which. If not, why not? 

Case 2:
Why is there no MRI scan? A pacemaker is not a contraindication (N Engl J Med 2017; 377:2555-2564; DOI: 10.1056/NEJMoa1604267).
Please explain all abbreviations at their first appearance (PACU, ASA).

Discussion:

No endovascular therapy is mentioned, please discuss the possibility of that. For example, see Shafafy et al. 2017 (J Spine Surg. 2017 Jun; 3(2): 217–225. doi: 10.21037/jss.2017.05.10).

There is much more literature you should discuss: A systematic review by Guan et al. 2017 (PMID: 28712898; DOI: 10.1016/j.wneu.2017.07.027).

Additionally Bonney et al. 2017: Vertebral artery injury in patients with isolated transverse process fractures (PMID: 28318982 DOI: 10.1016/j.jocn.2017.02.045)

Author Response

Point 1: Please start with a more general approach, i.e. on cervical surgery. Afterwards, the mentioned spectrum of complications can be elucidated.

Can the risk of injury of vertebral artery be assessed before the operation (s. Kong et al. 2019: PMID: 31428861; DOI: 10.1007/s00586-019-06111-0).

Motor strength was 4 out of 5": Please report the muscle, i.e. M. biceps or triceps brachii, or both.

Response to Point 1: We changed the manuscript to address these concerns. Also, in line 114, the muscle is reported.

Point 2: Please state votum of local ethics committee.

Response to Point 2: We are unsure of what is meant by ethics committee.  There is no ethics issue in these case reports.  Complications are routinely reviewed by our institutional Quality improvement committee and no negative qualifiers were documented. 

Point 3: Case 1: Surgical plan: 
Did you use intraoperative navigation systems? If so, which. If not, why not?

Response to Point 3: In lines 286-289, we made changes to address this in the manuscript.

Point 4: Case 2:
Why is there no MRI scan? A pacemaker is not a contraindication (N Engl J Med 2017; 377:2555-2564; DOI: 10.1056/NEJMoa1604267).

Response to Point 4: We made manuscript changes and addressed this in line 131 of the manuscript.

Point 5: Please explain all abbreviations at their first appearance (PACU, ASA).

Response to Point 5: Line 149, this change was made in the manuscript.

Point 6: No endovascular therapy is mentioned, please discuss the possibility of that. For example, see Shafafy et al. 2017 (J Spine Surg. 2017 Jun; 3(2): 217–225. doi: 10.21037/jss.2017.05.10).

Response to Point 6: In the manuscript we made changes in lines 200-204, we addressed endovascular therapy.

Point 7: There is much more literature you should discuss: A systematic review by Guan et al. 2017 (PMID: 28712898; DOI: 10.1016/j.wneu.2017.07.027).

Additionally Bonney et al. 2017: Vertebral artery injury in patients with isolated transverse process fractures (PMID: 28318982 DOI: 10.1016/j.jocn.2017.02.045)

Response to Point 7: In lines 187-205 we made changes to the manuscript and discussed the literature including the articles pertaining to Vertebral artery injury in cervical surgery cases. (Lunardini et al, Guan et al, Bonney et al ( in references).

Round 2

Reviewer 2 Report

All my concerns have been addressed.

The current manuscript is in a suitable form for publication.